# Measurements of Excavation Damaged Zone by Using Fiber Bragg Grating Stress Sensors

**DOI:** 10.3390/s21155008

**Published:** 2021-07-23

**Authors:** Xiaorong Wan, Chuan Li, Zhengang Zhao, Dacheng Zhang, Yingna Li, Jiahong Zhang

**Affiliations:** 1Faculty of Information Engineering and Automation, Kunming University of Science and Technology, Kunming 650500, China; xiaorong.wan@kust.edu.cn (X.W.); zhaozhengang@kust.edu.cn (Z.Z.); dacheng.zhang@kust.edu.cn (D.Z.); liyingna@kust.edu.cn (Y.L.); zhangjiahong@kust.edu.cn (J.Z.); 2Yunnan Key Laboratory of Computer Technology Applications, Kunming 650500, China

**Keywords:** coal and rock, excavation damaged zone, stress, differential fiber Bragg grating stress sensor, blasting excavation

## Abstract

In this paper, a Fiber Bragg Grating (FBG) stress sensor is developed to measure the stress variation between the lower Excavation Damaged Zone (EDZ) and the upper undistributed rock. The disturbance brought by the environmental temperature can be differentially compensated with two FBGs mounted symmetrically on the spokes. Through finite element analysis, it can be known that the direct stress and shear stress are pointed at the angles of 45° and 60° on both sides of the coal mine roadway, respectively. The anchor ends of the sensors are installed into the upper undistributed rock and the bolt tails of the mine roadway with a depth of 700 m and fastened by nuts to secure the load sensing device on the surface of the rock. When the shallow foundation of surrounding rock is pressed and deformed toward the coal mining road, the structural modifications can be converted into the stress of rock bolt and the strain of spoke. Thus, the FBG mounted on the surface of the spoke receives the shift information of the Bragg wavelength. The monitoring results indicate that the FBG stress sensors are sensitive to the variation of the EDZ. During the blasting, the stress amplitude varies from 40.256 to 175.058 kPa, and the creep time changes from 21 to 74 min. The proposed method can be applied in the field of underground coal mines for safety condition monitoring of the EDZ and forecasting the coal mine roadway stability.

## 1. Introduction

In the excavation of the coal mine roadway, the mining operation accelerates the deformation of the roadway and even causes disasters such as collapse [1,2]. According to the statistics, the number of casualties in coal mines due to roadway roof collapse in the past years accounts for 35–40% of the total number of all the accidents in coal mining. Therefore, the real-time stress monitoring of the coal mine roadway surrounding rocks and the corresponding security preventions are of great importance to avoid roof instability and sudden damages [3,4] However, the electronic sensors cannot satisfy the environmental conditions of coal mine roadway, such as darkness, damp, vibrations, electromagnetic interference, and flammable and explosive gases [5]. It is urgent to develop a monitoring system that meets the working conditions of coal mine tunnels and transmits the operation status of tunnels to the ground data center. Effective coal mine roadway monitoring can provide adequate warning to underground personnel before any unexpected major coal mine failure [6].

Based on the development of all-dielectric materials, FBG sensors have the advantages of corrosion resistance, electromagnetic interference resistance, and electrical insulation [7,8,9]. They can thus thrive in extreme underground environments, such that remote online monitoring can be realized. Li et al. [10] used an encapsulated FBG sensor and a distributed Brillouin tight buffered to measure the stress and strain of the tunnel lining. Zhu et al. [11] acquired the local pressure by using FBG pressure sensors. Most fiber-optic sensors are used to measure the local deformation. Gage et al. [12] used FBG strain sensors to obtain the strain of rock mass. By combining FBG strain sensors and Brillouin distributed fiber sensors, Pei et al. [13] pointed out that, based on the actual strain measured by the fiber-optic sensor, the transfer of various analysis methods to the required parameters such as displacement, force, and pressure can more directly reflect the safety of the rock and soil structure under complex engineering stress conditions. Zhao et al. [14] designed an FBG displacement sensor to eliminate the cross-sensitivity of temperature and displacement of FBG and prevent coal mine tunnel collapse by monitoring the roof displacement of the coal mine tunnel. Zhao et al. [15] used the FBG displacement sensor to measure the displacement of the roof separation during the blasting process. Zhao et al. [16] developed the FBG roof separation sensor and FBG stress sensor and formed a safety control monitoring system, which accurately captured the dynamic impacts of face advancement on roof displacement and stress. Jo et al. [17] researched a comprehensive mine structural safety system integrating high-precision FBG and obtained the influence of dynamic continuous coal mining on the stability of mine roadways and passages. Qin et al. [18] fabricated a novel FBG-based effective stress cell for directly measuring effective stress in saturated soils. The good performance of the FBG-based effective stress cell in saturated CDG soil under a multi-stage loading has been validated by comparison with the calculated results. Ren et al. [19] implemented the multi-point monitoring system based on FBG sensors array into the deep unconsolidated layer to realize a long-term ground monitoring, which investigated the compression process of ground rock layers at different burial depths and revealed the mechanical weakening characteristics of clay.

In this paper, a spoke-type FBG tension sensor based on mining bolts is developed, which can obtain the relative deformation of different layers (that is, the upper undistributed rock and the lower EDZ). The developed intrinsically safe fiber grating sensor is installed in the flammable and explosive coal mine yard 700 m underground, and the measured data are transmitted to the underground detection room through fiber-optic transmission. The fiber grating detection system is installed through the flameproof method. Finally, underground fiber-optic Ethernet is used to realize remote online monitoring of coal and rock pressure. The online monitoring data are transmitted through the underground fiber-optic Ethernet to the ground remote monitoring center and reflect the roadway stability status.

The remainder of this paper is organized as follows. In Section 2, the operation principle of the differential FBG stress sensor is explained. The online monitoring of stress loading in underground coal mine roadway is described in Section 3. The results are presented and discussed in Section 4.

## 2. Operation Principle of the Differential FBG Stress Sensor

Due to the great depth of the coal mine roadway and the low strength of the surrounding rock, the support strength of the roadway after excavation becomes too small compared with the ground stress, and it always has a time lag and lacks stickiness. Once the tolerance threshold of the surrounding rock is exceeded, the rock will be destroyed from the EDZ [1,2,3,20]. Therefore, the EDZ is developed by the comprehensive effects related to the surrounding rock stress, section span, shape, and other factors. All those factors compose a comprehensive index reflecting the surrounding rock stress and rock mass strength. A bolt-based spoke-type FBG stress sensor is developed to measure the deformation of the lower EDZ and the upper undistributed rock.

As shown in Figure 1, the FBG stress sensor consists of a bolt implanted in the roof rock, a spoke-type force measuring structure joined to the bolt, and a pallet fastened on the surface of the roof rock. The excavation failure area was penetrated by a 6 m anchor rod. The roof-bolt head is anchored in the upper unexcavated area. The sensor is fixed on the surface of the excavation area. The stress of the entire excavation damage zone is transmitted to the strip fiber grating stress sensor, which realizes the equivalent stress monitoring of the entire rock formation in the failure zone along the bolt direction. As the bolt is subjected to respond to the change between the lower EDZ and the upper undistributed rock, the spoke-type force measuring structure introduces the shearing deformation of the spoke under the reaction force of the pallet. Two FBGs are mounted symmetrically along the neutral axis with an angle of ±45${}^{\circ }$ from their central position. The bolt passes through the hub with a diameter of 25 mm. As the hub is subjected to the shearing forces derived from the applied force of the rock bolt, four spokes are deformed. Whether the mounted grating is compressed or stretched, the response strain of wavelength shifts, denoted ε, and the temperature change ΔT can be expressed as [7,21,22]:(1)$$\frac{\Delta \lambda_{B}}{\lambda_{B}}=\left(1-p_{e}\right)\varepsilon +\left(\alpha_{n}+\alpha_{\Lambda }\right)\Delta T+\left(1-p_{e}\right)\left(\alpha_{H}-\alpha_{\Lambda }\right)\Delta T$$
where $\lambda_{B}$ is the Bragg wave length, $\Delta \lambda_{B}$ is the shift of Bragg wavelength, *p_e_* = 0.22 is the effective elastic-optic coefficient for the fiber, $\alpha_{n}=6.12\times 10^{-6}$, ${}^{\circ }C^{-1}$ is the thermo-optic coefficient, $\alpha_{\Lambda }=0.55\times 10^{-6}$${}^{\circ }C^{-1}$ is the thermal expansion coefficient, $\alpha_{H}=16.0\times 10^{-6}$ ${}^{\circ }C^{-1}$ is the thermal expansion coefficient for the spoke, and $S_{H}=\left(1-p_{e}\right)\left(\alpha_{H}-\alpha_{\Lambda }\right)$ is the additional strain sensitivity derived from the thermal expansion coefficient mismatch between the fiber and the spoke. Moreover, the strain and thermal coefficients of relative Bragg wavelength shifts $\lambda_{B}/\Delta \lambda_{B}$ are 0.78 × $10^{-6}$
$\mu \varepsilon^{-1}$ and 18.72 × $10^{-6}$
${}^{\circ }C^{-1}$, respectively. In Table 1, the bolt head is made of stainless steel with the elasticity modulus *E* = 214 GPa and the Poisson ratio *v* = 0.27. The thickness of rim is 25 mm with the reserved height of 5 mm below the hub for overload protection. The length of spoke is *l* = 22 mm, the height of spoke is *h* = 5 mm, the width of spoke is *b* = 1 mm, and the cross-sectional area *A* = *hb* is equal to 5 mm^2^. The diameter of bolt is *d* = 20 mm.

As the hub is subjected to an axial force *F*, the shearing force generated on the cross-section of each spoke is equal to *F*/4, and the local shear stress of spoke is formulated as [23,24,25]:(2)$$\tau =\frac{F(h^{2}/4-y^{2})}{2bh^{3}/3}$$
where *y* is the relative distance from the center of spoke. The shear stress $\tau_{max}$ at the neutral plane y=0 is expressed as:(3)$$\tau_{max}=\frac{3F}{8bh}$$

The normal stress $\left|\sigma \right|=\tau_{max}$ is maximal at the angle of ±45${}^{\circ }$ with the neutral plane. According to Hooke’s law, the primary strains $\varepsilon_{1}$ and $\varepsilon_{2}$ can be expressed by:(4)$$\left\{\begin{array}{c} \varepsilon_{1}=\frac{3(1+v)F}{8EA}\;\\ \varepsilon_{2}=-\frac{3(1+v)F}{8EA} \end{array}\right.$$

The response of FBGs to the external force *F* is given by substituting Equation (Equation 4) into Equation (Equation 1):(5)$$\begin{array}{c} \left\{\begin{array}{c} \frac{\Delta \lambda_{1}}{\lambda_{1}}=\left(1-p_{e}\right)\frac{3(1+v)}{8EA}F+\left(\alpha_{n}+\alpha_{\Lambda }\right)\Delta T+\left(1-p_{e}\right)\left(\alpha_{H}-\alpha_{\Lambda }\right)\Delta T\\ \frac{\Delta \lambda_{2}}{\lambda_{2}}=-\left(1-p_{e}\right)\frac{3(1+v)}{8EA}F+\left(\alpha_{n}+\alpha_{\Lambda }\right)\Delta T+\left(1-p_{e}\right)\left(\alpha_{H}-\alpha_{\Lambda }\right)\Delta T \end{array}\right. \end{array}$$
where $\lambda_{1}$ and $\lambda_{2}$ are the Bragg wavelength of gratings mounted on spoke along the neutral axis with the angles of ±45${}^{\circ }$ symmetrically. According to Equation (Equation 5), the disturbance from environmental temperature can be compensated by a differential operation:(6)$$\frac{\Delta \lambda_{1}}{\lambda_{1}}-\frac{\Delta \lambda_{2}}{\lambda_{2}}=\left(1-p_{e}\right)\frac{3(1+v)}{4EA}F$$
where $\Delta \lambda_{1}/\lambda_{1}-\Delta \lambda_{2}/\lambda_{2}$ is the difference of the relative change of the reflection wavelength of two FBGs. The pressure *F* was transmitted from the bolt with the diameter of *d* subjected by the coal and rock stress σ:(7)$$\sigma =\frac{4F}{\pi d^{2}}$$

Substituting Equation (Equation 7) into Equation (Equation 6), the theoretical sensitivity of differential FBG stress sensor is formulated by:(8)$$\frac{\Delta \lambda_{1}}{\lambda_{1}}-\frac{\Delta \lambda_{2}}{\lambda_{2}}=\left(1-p_{e}\right)\frac{3\pi (1+v)}{16EA}d^{2}\sigma =S_{Stress}\sigma$$
where $S_{Stress}=\left(1-p_{e}\right)\frac{3\pi (1+v)}{16EA}d^{2}$ is the theoretical sensitivity of differential FBG stress sensor. Plugging the correlation parameters of Table 1 into Equation (Equation 8), the theoretical sensitivity of differential FBG stress sensor is $2.181\times 10^{-10}$ Pa^−1^. According to the measurement principle of the FBG pressure sensor, it can be seen from Equation (Equation 8) that the influence of temperature on the grating is canceled out, and the output of the system is a function of strain.

As shown in Figure 2, the broadband light is injected into the FBGs via a coupler of 3 dB. Then, the lights reflected from the gratings are returned. The peak detection is employed to determine where the photo-detector receives the maximum back-reflected light from the FBG sensor. The wavelength measurement accuracy of this modulator is 0.001 nm. The two gratings of the sensor are completely symmetrical and have the same response characteristics to ambient temperature. During the test, the spoke-type FBG stress sensor is placed on the base of the pressure test machine. When the pressure is loaded and unloaded, the strain of mounted grating is changed with the bending moment of the spoke. After five repeated experiments, the sensor repeatability error is less than 1.2%, which meets the requirements of coal mine roadway stress monitoring. The change in the center wavelength of the reflected light of the two gratings with the applied load is shown in Figure 3. The results of the pressure test show that the measured sensitivity of differential FBG stress sensor is $2.344\times 10^{-10}$${\mathrm{Pa}}^{-1}$, with linearity of 0.23% at the measuring pressure in the range from 0 to 1000 N, i.e., the corresponding stress is 0∼3.183 ×$10^{6}$ Pa, as shown in Figure 4.

## 3. Online Monitoring of Stress Loading in Underground Coal Mine Roadway

The coal mine roadway excavation destructs the original static equivalence of the surrounding rock mass, resulting in the secondary stress distribution of the surrounding rock. If the secondary stress reached or exceeded the strength limit of the surrounding rock, the rock will be destroyed. Thus, the monitoring of the stress of the coal mine roadway can provide key state data for the deformation law of the surrounding rock. The mine area located in middle China was deformed severely during the periods 205–257 and 65–137 million years ago, which resulted in a multi-fracture zone. Since most of the coal mine roadways have been severely deformed, the production is significantly affected.

Stress values and directions vary considerably at different locations in the roadway cross-section. To find the best position for the sensor installation and obtain effective monitoring results, the stress field in the surrounding rock was analyzed through the finite element analysis. The roadway is located 700 m underground, and the roadway surface is covered with a lining structure with a thickness of 30 cm, and the surrounding of the lining layer is rock. The elastic modulus, Poisson’s ratio, and density of the lining layer are 5 GPa, 0.3, $1.8\times 10^{3}$
$\mathrm{kg}/m^{3}$, respectively, and of the rock are 32 GPa, 0.3, and 2.3 $\mathrm{kg}/m^{3}$, respectively. According to the structure parameters of the roadway model, Figure 5 and Figure 6 show the diagrams of direct stress fields and shearing stress fields of coal mine roadway before and after excavation, respectively.

In Figure 5 and Figure 6, the equilibrium status of the original stress field of the surrounding rock is broken after the excavation of the coal mine roadway. The surrounding rock stress field is thus redistributed. Due to the release of surrounding rock stress after excavation, the stress before excavation is greater than the stress after excavation, and the shear stress of the vault has not changed significantly. However, the direct stress and the shear stress are pointed at the angles of 45° and 60° on both sides of the coal mine roadway. Thus, four probing points are set with two different angles of 45° and 60° in the coal mining road. Holes are drilled with a diameter of 28 mm and a depth of 6 m. The GM22/2500-490 type mining bolts are placed on the borehole base. The bolts’ ends are fixed by using an anchoring agent, while the bolts pass through the pallet. The FBG sensor hub is fastened to the surrounding rock surface with four nuts, as shown in Figure 7. The ends of the bolts are fixed in the upper undistributed rock. The pallet is subjected to the extrusion deformation of the coal mine roadway induced by the unstable rock. The FBG stress sensors are installed between the pallet and the fastening nut. The FBG tension sensor transmits the deformation of the coal seam in an axial force, resulting in a corresponding Bragg wavelength shift.

The bolt-based FBG stress sensors installed in the designed position do not require electrical power since the FBG sensors are connected by fiber-optics. The FBG monitoring system is placed in the underground monitoring cabin. In order to adapt to the underground coal environment, the system is installed in an explosion-proof box. A 127 V power supply is provided to the fiber Bragg grating monitoring system via a coil safety cable. The monitoring system consists of an amplified spontaneous emission (ASE) light source, an interrogation analysis, and peripheral circuitry. The broadband light (optical spectrum from 1525 nm to 1565 nm) by the ASE is injected into the FBG sensor through the fiber monitoring net, and the sensor grating returns the optical signal of a specific wavelength, with the incident and reflected beams sharing the same physical path through a 3 dB fiber coupler. The reflected light enters the interrogation analysis, which detects and outputs the central wavelength of the maximum back-reflected light. In the monitoring system, the Bragg wavelength shift of FBGs is received and transformed into stress variations. The collected data are transmitted through the underground fiber-optic Ethernet to the ground remote monitoring center, as shown in Figure 8.

## 4. Results and Discussion

The test roadway is a working coal mine. The installed sensors are 150 m away from the mining face. In the long-term monitoring of the roadway, the key data of the roadway stress before and after blasting and the impact of blasting on the roadway stress can be obtained. During the monitoring period of 9600 min, the stress variations were monitored by the sensors. There were three blastings per delay during the monitoring time in the coal mine and three stress mutations between the lower EDZ and the upper undistributed rock, accordingly. The maximum stress increment during the excavation blasting was 175.058 kPa.

In order to compare the sudden variations in the rock stress during the three blasting, the stress variations during the blasting monitored by the sensors are shown in Figure 9. Affected by the blasting, the variation trends of the stresses are exactly the same. The peak values of stress variations during excavation blasting are shown in Table 2. The first blasting brings the larger stress variation and the second blasting causes the smaller one. The stress variations during three blastings are caused by the amount of blasting explosives per delay. The creeping time (includes the ascent time and the recovery time) of three blastings are about 21–74 min. Furthermore, the larger the stress varied, the more the creep time cost. The creep times are approximately proportional to the stress peak values.

The monitoring results indicate that the FBG stress sensor installed on the roadway surface can be used for EDZ monitoring in a coal mine roadway. By analyzing the monitoring data, when the stress is greater than the set threshold, the system can give an alarm signal in time. Thus, this method can be used to monitor the stress of the undistributed rock strata and the lower EDZ in the process of underground mining and provide the key decision data for safe production.

## 5. Conclusions

In underground engineering and mining engineering, the surrounding rock may be deformed, broken, or even unstable due to excavation and blasting. Therefore, the real-time monitoring of surrounding rock is particularly important to the safety of mine roadways. In this work, an FBG stress sensor was developed to monitor the EDZ of the coal mine roadway. By using the differential operation, the disturbance of environmental temperature can be eliminated effectively. In the testing of the coal mine roadway, the FBG sensors were installed directly into the flammable and explosive coal mine roadway, whereas the FBG monitoring system was placed in the underground monitoring cabin. The online monitoring data were transmitted through the underground fiber-optic Ethernet to the ground remote monitoring center. It is noteworthy that the sensors installed on the roof and sides of the roadway revealed the stress variation in the blasting process. 

## Figures and Tables

**Figure 1 sensors-21-05008-f001:**
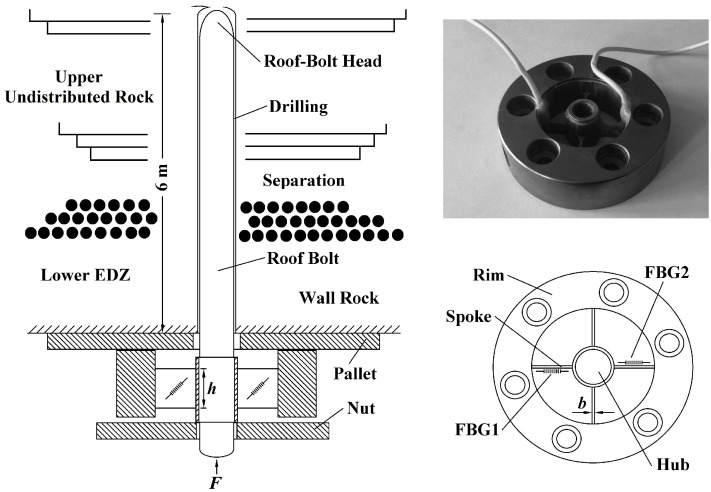
Schematic digram of spoke-type fiber Bragg grating stress sensor based on mine bolt.

**Figure 2 sensors-21-05008-f002:**
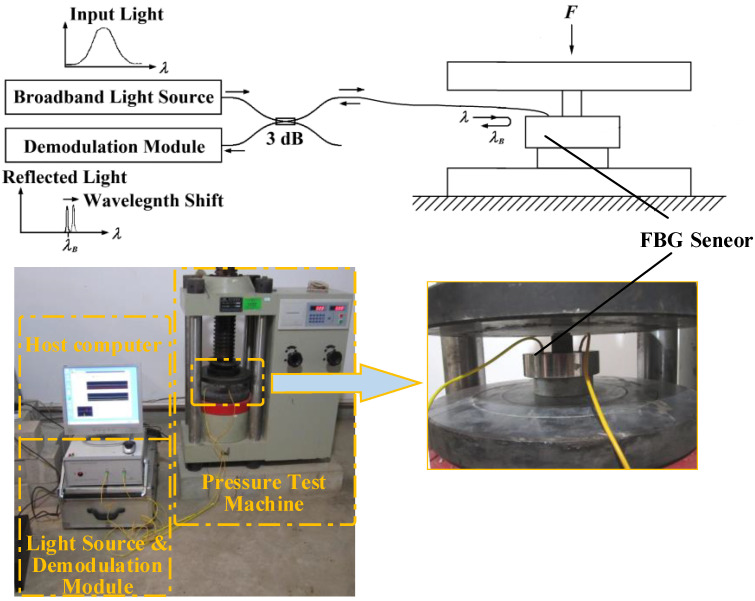
Pressure testing digram of spoke-type fiber Bragg grating stress sensor.

**Figure 3 sensors-21-05008-f003:**
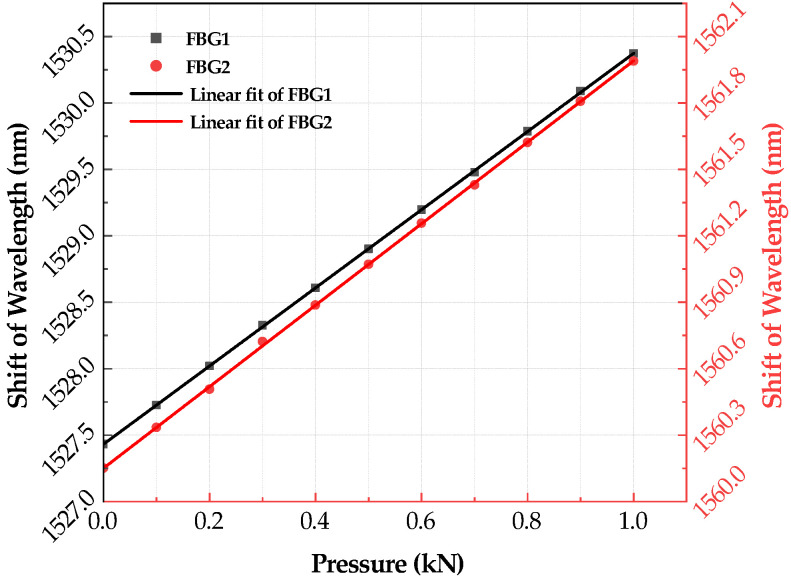
The test data of spoke-type fiber Bragg grating stress sensor.

**Figure 4 sensors-21-05008-f004:**
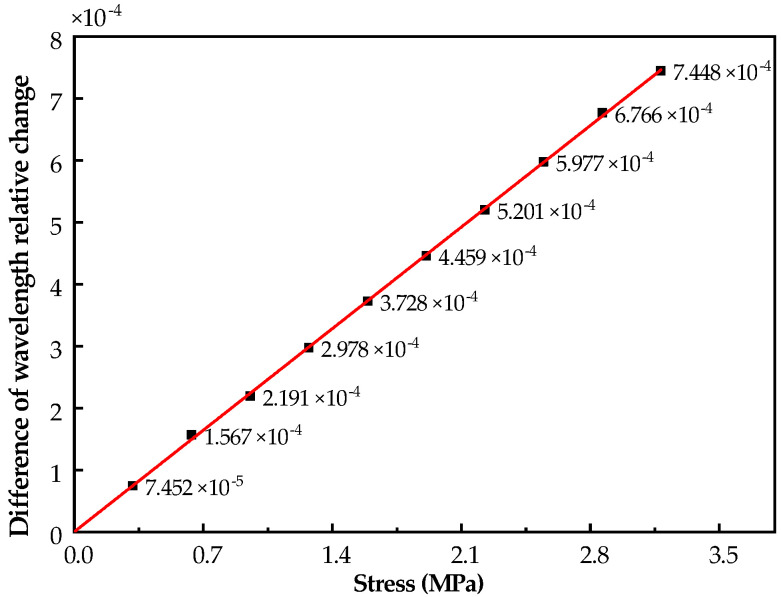
Difference of wavelength relative change within the measurement range.

**Figure 5 sensors-21-05008-f005:**
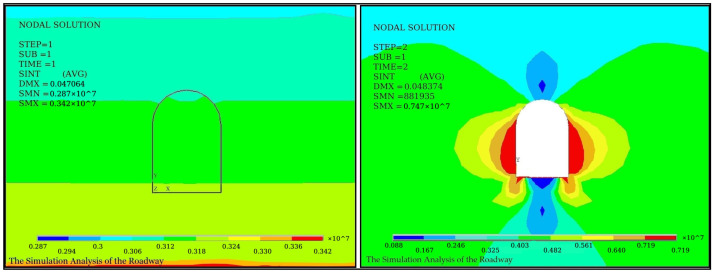
Diagram of stress fields before and after excavation.

**Figure 6 sensors-21-05008-f006:**
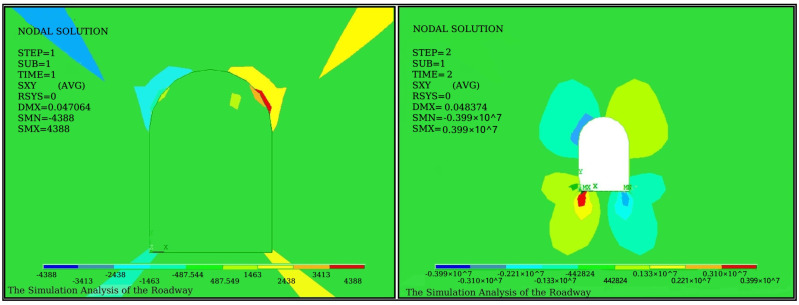
Diagram of shear stress fields before and after excavation.

**Figure 7 sensors-21-05008-f007:**
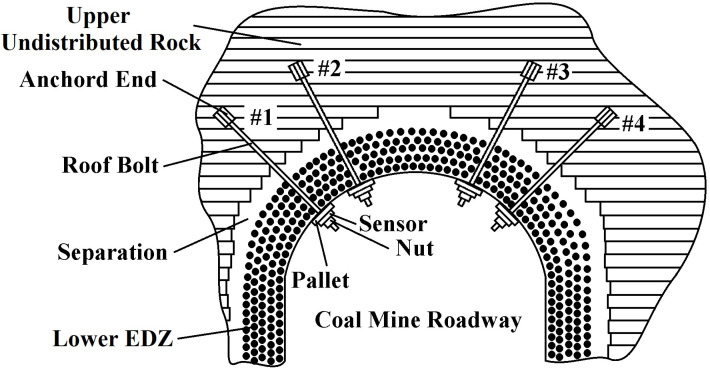
Monitoring schematic diagram of fiber Bragg grating stress sensors installed on the roof and sides of the roadway.

**Figure 8 sensors-21-05008-f008:**
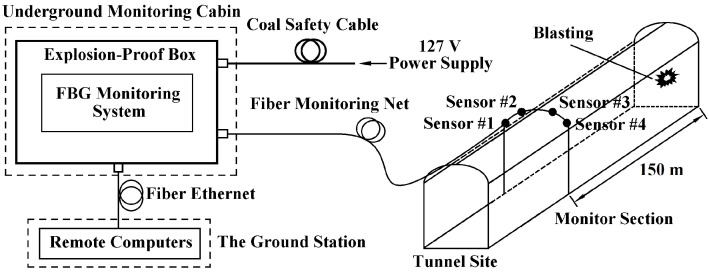
Underground fiber Bragg grating remote monitoring system.

**Figure 9 sensors-21-05008-f009:**
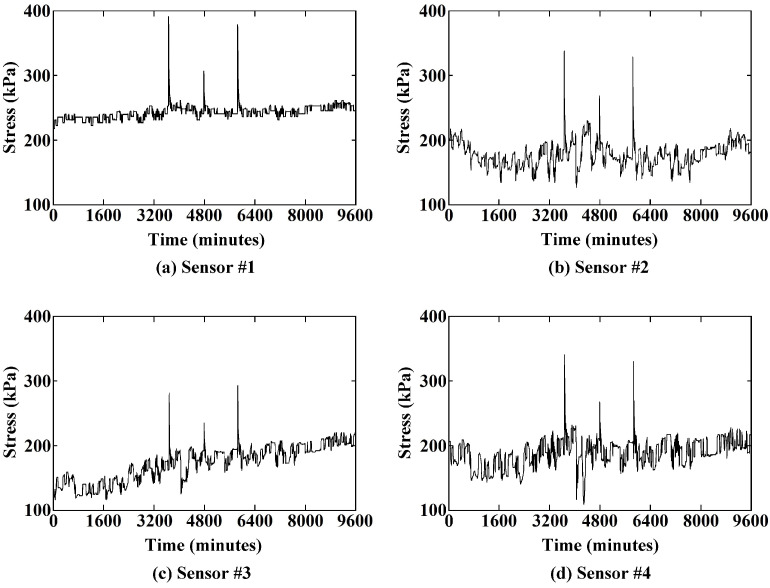
Time-varying diagram of stress for characterizing shallow creep of the coal mine roadway.

**Table 1 sensors-21-05008-t001:** The structural parameters of fiber Bragg grating stress sensor.

Parameter	Notation	Value
elasticity modulus	*E*	214 GPa
Poisson ratio	*v*	0.27
the length of spoke	*L*	22 mm
the height of spoke	*H*	5 mm
the width of spoke	*B*	1 mm
the cross-sectional area	A=hb	5 ${\mathrm{mm}}^{2}$
the diameter of bolt	*d*	20 mm

**Table 2 sensors-21-05008-t002:** Characteristics of stress during blasting recorded by sensors.

Blasting		Sensor #1	Sensor #2	Sensor #3	Sensor #4
First blasting:	Stress Variation (kPa)	135.969	175.058	113.041	155.077
3307th minute	Creep (min)	54	65	48	74
Second blasting:	Stress Variation (kPa)	64.853	93.715	40.256	72.842
4291st minute	Creep (min)	26	41	21	35
Third blasting:	Stress Variation (kPa)	125.900	155.192	106.554	150.927
5190th minute	Creep (min)	53	45	48	41

## Data Availability

The data that support the findings of this study are available from the corresponding author upon reasonable request.

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
