# Peer review of "Measurements of Excavation Damaged Zone by Using Fiber Bragg Grating Stress Sensors"

_sensors, 2021, doi:10.3390/s21155008_

Round 1
Reviewer 1 Report
Authors presented the application of FBG sensors on the stress variation assessment in excavation damage zones. Although the paper presents a practical application of optical fiber sensor, it needs many improvements on the methodology and presentation quality. Thus, it is suggested that the authors consider the comments presented below to thoroughly improve the paper before a final decision.
- Introduction is too brief and does not include a broad overview of the theme. The authors should further discuss the motivations and the contributions of this work should be included. The discussion of contributions and novelty of the work are key topics in introduction sections, but both of them are missing in this work.
- In general, the references are not updated, the vast majority of the works are from more than 3 years ago, especially the references related to the FBG sensors. Authors should update their references with the current state of the art, FBGs (and optical fiber sensors) are a widespread technology and there are dozens of recent works.
- The photographs of Figure 2 should be improved, the authors should indicate the key elements of the figure. In addition, the black and white figure harms the visualization. Authors should consider adding a colored figure, since black and white figures are not mandatory in this journal.
- Figures 3 and 4 are not visible at all. Once again the black and white display harms the visualization, but, in this case, there is also the small fonts. In addition, the software is spelled wrong in the abstract.
- The experimental results are only shown briefly in figure 7. Authors mentioned a pressing test on the sensor, but the results were never shown. In addition, the authors mentioned a temperature compensation technique only in the abstract with no results shown. Authors should include at least the pressure (or stress) characterization tests of the sensor for the evaluation of the wavelength shift as a function of the applied stress. Furthermore, the performance of the temperature compensation using dual FBGs approach should be investigated with temperature characterization tests.
Author Response
Thank you for your comments concerning our manuscript entitled Measurements of Excavation Damaged Zone by Using Fiber Bragg Grating Stress Sensors (Manuscript ID: sensors-1278174). Those comments are all valuable and very helpful for revising and improving our paper, as well as the important guiding significance to our researches. We have studied comments carefully and have made correction which we hope meet with approval. The modifications are highlighted in red in the revised manuscript. The main corrections in the paper and the responds to the reviewer’s comments are as follows:
- Introduction is too brief and does not include a broad overview of the theme. The authors should further discuss the motivations and the contributions of this work should be included. The discussion of contributions and novelty of the work are key topics in introduction sections, but both of them are missing in this work.
Reply: We have added the discussion and contribution into the Introduction section.
- In general, the references are not updated, the vast majority of the works are from more than 3 years ago, especially the references related to the FBG sensors. Authors should update their references with the current state of the art, FBGs (and optical fiber sensors) are a widespread technology and there are dozens of recent works.
Reply: The references have been updated to the current state of the art works, references [4], [5], [6], [17], [18] , and [19].
- The photographs of Figure 2 should be improved, the authors should indicate the key elements of the figure. In addition, the black and white figure harms the visualization. Authors should consider adding a colored figure, since black and white figures are not mandatory in this journal.
Reply: Figure. 2 has been modified according to the suggestions.
- Figures 3 and 4 are not visible at all. Once again the black and white display harms the visualization, but, in this case, there is also the small fonts. In addition, the software is spelled wrong in the abstract.
Reply: Figures 3 and 4 have been modified and the Abstract section has been paraphrased.
- The experimental results are only shown briefly in figure 7. Authors mentioned a pressing test on the sensor, but the results were never shown. In addition, the authors mentioned a temperature compensation technique only in the abstract with no results shown. Authors should include at least the pressure (or stress) characterization tests of the sensor for the evaluation of the wavelength shift as a function of the applied stress, Furthermore, the performance of the temperature compensation using dual FBGs approach should be investigated with temperature characterization tests.
Reply: The stress test results of the fiber grating stress sensor is added in the original text (Figure 3 and 135 lines). Combining the test principle (Equation 6) and the drawn figure 4, the origin of the measurement sensitivity and linearity index of the differential sensor is explained. The temperature test method and test results of the fiber grating stress sensor are added to the original text, and the temperature compensation results are described in the original text.
Reviewer 2 Report
The authors demonstrate a Measurements of excavation damaged zone by using fiber Bragg grating stress sensors. Some of the simulation and experimental results are interesting. However, there are some points should be emphasized and interpreted. Therefore, the following comments may help the authors to modify the manuscript:
- Authors may need to improve the English presentation of the manuscript.
- The authors should clarify the advantages of this structure over other similar works.
- Authors should compare their simulation results with those reported in the literature in a table
- Based on the obtained results, it is really not impressive, and most of the commercial product able to function better than this. Authors should highlight the novelty of the obtained results, particularly in the potential application.
- What is the repeatability of the measurements?
Author Response
Thank you for your comments concerning our manuscript entitled Measurements of Excavation Damaged Zone by Using Fiber Bragg Grating Stress Sensors (Manuscript ID: sensors-1278174). Those comments are all valuable and very helpful for revising and improving our paper, as well as the important guiding significance to our researches. We have studied comments carefully and have made correction which we hope meet with approval. The modifications are highlighted in red in the revised manuscript. The main corrections in the paper and the responds to the reviewer’s comments are as follows:
- Authors may need to improve the English presentation of the manuscript.
Reply: We have checked the whole text regarding grammar and expressions.
- The authors should clarify the advantages of this structure over other similar works.
Reply: The excavation failure area was penetrated by a 6-meter-long anchor rod. The Roof-bolt Head is anchored in the upper unexcavated area. The sensor is fixed on the surface of the excavation area. The stress of the entire excavation damage zone is transmitted to the strip fiber grating stress sensor, which realizes the equivalent stress monitoring of the entire rock formation in the failure zone along the bolt direction. (line 87-92, marked in red).
The two gratings of the sensor are completely symmetrical, and have the same response characteristics to ambient temperature. (line 138, marked in red). According to the measurement principle of the FBG pressure sensor, it can be seen from formula 8 that the influence of temperature on the grating is cancelled out, and the output of the system is a function of strain. (line 134, marked in red).
- Authors should compare their simulation results with those reported in the literature in a table.
Reply: In the simulation model, it is assumed that the rocks around the excavation area are evenly distributed and the material properties are the same, which are simulation results under ideal conditions. It can only qualitatively describe the stress distribution of Excavation Damaged Zone in a larger area, and provide guidance for the selection of sensor placement.
However, in the actual environment, the distribution of rocks and coal seams is more variable, and the material properties are very complex, and the measurement data can only reflect the stress of the local measurement points. Therefore, there is a certain difference between the simulation result and the actual measurement result. According to the actual situation, the author omitted the comparison between the simulation result and the monitoring result.
- Based on the obtained results, it is really not impressive, and most of the commercial product able to function better than this. Authors should highlight the novelty of the obtained results, particularly in the potential application.
Reply: The fiber Bragg grating tension sensor based on the mine bolt, the bolt is 6 meters long, and can be used for overall monitoring from the deep coal seam to the surrounding rock of the roadway. Compared with general surface-attached strain sensors, the measurement range is deeper and can reflect the overall stress concentration in the lower excavation area. In addition, the differential structure can better shield the temperature interference and more accurately reflect the stress situation. It also has good engineering application prospects and commercial value.
- What is the repeatability of the measurements?
Reply: Before the sensor was installed on the site, we calibrated the sensor many times to meet the repeatability measurement requirements. It can also be seen from the three blasting monitorings in the field monitoring that all four sensors can track the loading and unloading changes of the blasting stress well, and have good measurement repeatability. (line 145, marked in red)
Reviewer 3 Report
Bragg gratings are known to detect deformations operating on an area of 1cm2 around their locations in optical fibers. Therefore, it will be necessary to explain how we could justify deformations on a much larger surface… It would not only be necessary to place a very large number of FBGs on a surface but also to bury them at several depths to be sure of the result. Moreover, how can we be sure that the graphs in figure 7 are indeed perennial deformations and not elastic due to explosions? Finally, the results part must absolutely include comparisons with other work using other techniques in order to prove the usefulness of the
Author Response
Thank you for your comments concerning our manuscript entitled Measurements of Excavation Damaged Zone by Using Fiber Bragg Grating Stress Sensors (Manuscript ID: sensors-1278174). Those comments are all valuable and very helpful for revising and improving our paper, as well as the important guiding significance to our researches. We have studied comments carefully and have made correction which we hope meet with approval. The modifications are highlighted in red in the revised manuscript. The main corrections in the paper and the responds to the reviewer’s comments are as follows:
1. Bragg gratings are known to detect deformations operating on an area of 1cm2 around their locations in optical fibers. Therefore, it will be necessary to explain how we could justify deformations on a much larger surface. It would not only be necessary to place a very large number of FBGs on a surface but also to bury them at several depths to be sure of the result.
Reply: The fiber Bragg grating tension sensor based on the mine bolt, the bolt is 6 meters long, and can be used for overall monitoring from the deep coal seam to the surrounding rock of the roadway. Compared with general surface-attached strain sensors, the measurement depth is deeper and can reflect the overall stress concentration in the lower excavation area.
2. Moreover, how can we be sure that the graphs in figure 7 are indeed perennial deformations and not elastic due to explosions?
Reply: From the monitoring data, it can be seen that the fluctuation of the smaller strain value is the strain generated by rock creep in the unstable phase of the lower excavation area, while three larger strain pulses were monitored by multiple sensors at the same time, and the time of occurrence of the large strain was consistently matched with the blasting time. Therefore, it is determined that the smaller strain fluctuations are caused by perennial deformation and the large strain abrupt changes are caused by blasting.
3. Finally, the results part must absolutely include comparisons with other work using other techniques in order to prove the usefulness of the
Reply: Considering the construction cost, the construction environment ,and the intrinsic safety of the sensors, we installed only four FGB sensors in the coal mine tunnel and did not compare other types of sensors for verification. Moreover, the monitoring of the overall strain in the lower excavation area of the coal mine has not been carried out before, so there is no comparative data. However, the stress results monitored by the four sensors are consistent and can reflect the effectiveness of the monitoring.
Round 2
Reviewer 1 Report
Authors addressed my comments and suggestions.